# Task-customized Masked AutoEncoder via Mixture of Cluster-conditional Experts

**Zhili Liu**[1,2], **Kai Chen**[1], **Jianhua Han**[2], **Lanqing Hong**[2], **Hang Xu**[2],
**Zhenguo Li**[2], **James T. Kwok** [1]

[1] Department of Computer Science and Engineering, Hong Kong University of Science and Technology
[2] Huawei Noah's Ark Lab

```
{zhili.liu, kai.chen}@connect.ust.hk,
{hanjianhua4, honglanqing, xu.hang, li.zhenguo}@huawei.com
jamesk@cse.ust.hk
```

## Abstract

Masked Autoencoder (MAE) is a prevailing self-supervised learning method that achieves promising results in model pre-training. However, when the various downstream tasks have data distributions different from the pre-training data, the semantically irrelevant pre-training information might result in negative transfer, impeding MAE's scalability. To address this issue, we propose a novel MAE-based pre-training paradigm, Mixture of Cluster-conditional Experts (MoCE), which can be trained once but provides customized pre-training models for diverse downstream tasks. Different from the mixture of experts (MoE), our MoCE trains each expert only with semantically relevant images by using cluster-conditional gates. Thus, each downstream task can be allocated to its customized model pre-trained with data most similar to the downstream data. Experiments on a collection of 11 downstream tasks show that MoCE outperforms the vanilla MAE by 2.45% on average. It also obtains new state-of-the-art self-supervised learning results on detection and segmentation.

## 1 Introduction

Self-supervised learning (SSL), which learns effective transferable representations without human annotations, has become a prevailing model pre-training paradigm (He et al., 2020; Chen et al., 2021a; Bao et al., 2022). Currently, the most prevalent SSL method is the Masked Autoencoder (MAE) (He et al., 2022), which constructs supervision signals from raw image data by masking random input patches and then reconstructing the missing pixels. This simple strategy has proved efficient in the training of large-scale models. For example, ViT (Dosovitskiy et al., 2021) shows impressive performance on popular benchmarks such as the ImageNet [1] (Deng et al., 2009). However, does MAE really scale well for various downstream tasks (Deng et al., 2009; Lin et al., 2014; Zhou et al., 2019; Han et al., 2021; Li et al., 2022a)?

Preliminary studies (in Section 3.1) show that the MAE indeed suffers from *negative transfer* (Liu et al., 2022) when transferring to downstream tasks with very different semantics. Figure 1(a) shows that on 9 of 11 downstream tasks, an MAE pre-trained on the full ImageNet data is outperformed by the one that is pre-trained on only the semantically relevant data subsets. Hence, using pre-training data that are semantically irrelevant can hurt transfer performance.

The above observation motivates the need for task-customized pre-training. A promising model for this is the Mixture of Experts (MoE) (Shazeer et al., 2017; Riquelme et al., 2021), which uses a multi-expert architecture to provide customized models for different input tokens. However, unlike supervised pre-training, self-supervised learning lacks semantic labels, and thus the experts differ more on low-level information than semantics, referring to Figure 1(b). Experiments in Section 4.2 show that a naive adoption of MoE to the MAE has inferior performance. Since various downstream tasks contain different semantics, semantic-related experts may be preferred.

---

[1] We refer to ImageNet-1K as ImageNet if not specified in this paper.

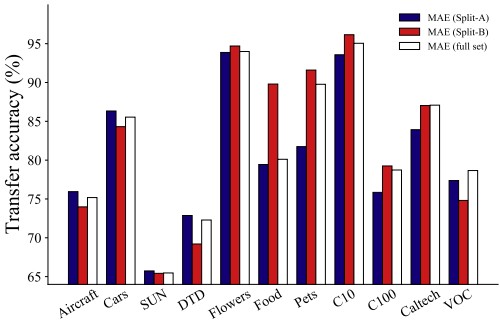

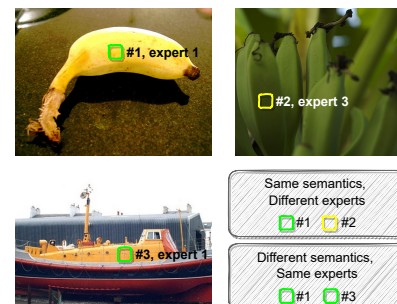

(a) Negative transfer phenomenon on MAE.

(b) Problem with TokenMoE.

Figure 1: (a) Transfer performance of MAEs pre-trained on Split-A (blue), Split-B (red) and full ImageNet data (white). Only two of the eleven downstream tasks benefit from using the full ImageNet data for pre-training (more details in Section 3.1). (b) TokenMoE uses pixel RGB values as reconstruction targets. Thus, tokens with similar pixel values tend to be routed to the same expert, leading to two types of mistakes: (i) same semantics but routed to different experts, (ii) different semantics but routed to the same expert.

In this paper, we propose the Mixture of Cluster-conditional Expert (MoCE), a novel paradigm to achieve task-customized self-supervised pre-training by data clustering and explicitly training each expert with images of similar semantics. The MoCE procedure has three stages. First, we cluster the whole dataset by using a pre-trained, dense MAE model. We then construct the MoCE with a multi-expert structure. Each expert is trained using clusters selected by routing tokens based on *cluster embedding* (instead of *token embedding*). To stabilize training and enhance confidence of the gate results, a regularization loss is proposed. Finally, with the arrival of a downstream task, we propose a search procedure to select the closest cluster. Empirically, the proposed MoCE shows superior performance over MAE on a collection of 11 downstream tasks. Besides, one can use only a MoCE sub-model on deployment, thus saving inference time and model capacity.

To summarize, our main contributions are:

1. We systematically analyze the negative transfer phenomenon of MAE, and show that naively adopting the MoE to MAE cannot improve transfer performance of downstream tasks.

2. We propose the MoCE, which trains each expert with semantics-aware clusters so that similar clusters can be routed to the same expert.

3. We demonstrate effectiveness of the proposed MoCE on a collection of 11 downstream tasks, and achieve up to 2.45% performance improvement in Top-1 accuracy. State-of-the-art self-supervised results are also achieved on the detection and segmentation tasks. To the best of our knowledge, this is the first work that achieves state-of-the-art transfer performance by training vision MoE models with ImageNet under the SSL setting.

## 2 RELATED WORK

**Self-supervised Learning.** Previous works mainly focus on the design of pretext tasks with image transformations (Doersch et al., 2015; Gidaris et al., 2018), inpainting (Pathak et al., 2016), colorization (Zhang et al., 2016), contrastive learning (Chen et al., 2020; He et al., 2020; Grill et al., 2020; Caron et al., 2020; Radford et al., 2021b; Yao et al., 2022b), and for specific downstream tasks (Wang et al., 2020; Xie et al., 2020; 2021a; Chen et al., 2021a; Yao et al., 2022a). Motivated by the design of BERT (Devlin et al., 2018), masked image modeling (MIM) is recently proposed to learn by reconstructing masked images. BEiT (Bao et al., 2022) is the pioneering work that predicts visual tokens generated by a pre-trained tokenizor (Radford et al., 2021a). SimMIM (Xie et al., 2021c) simplifies the framework by directly utilizing the pixel RGB values as reconstruction targets.

MAE (He et al., 2022) proposes an asymmetric encoder-decoder architecture for better training efficiency. MixedAE (Chen et al., 2023) further explores image mixing for object-aware pre-training. In this paper, we will focus on the MAE due to its effectiveness and efficiency.

While self-supervised learning methods have achieved improved transfer performance, most of them only provide a unified representation to various downstream tasks. This may suffer from negative transfer as demonstrated in Section 3.1. The work most relevant to ours is SDR (Liu et al., 2022), which trains 256 subnets with 256 disjoint ImageNet subsets simultaneously. However, this paper differs from SDR in three ways: (i) the mapping from subsets to subnets in SDR is randomly selected and fixed during pre-training, while MoCE achieves self-adaptive mapping with cluster-conditional gates; (ii) Progressive training is required in SDR, while MoCE enjoys one-time end-to-end training; (iii) During the transfer process, SDR uses brute force to select the best sub-model, while MoCE reuses the clustering module to achieve more efficient selection.

**Mixture of Experts.**    The mixture of experts (MoE) has a long history (Jacobs et al., 1991; Jordan & Jacobs, 1994; Shazeer et al., 2017). Recently, it is considered as an effective tool for model scale-up in natural language processing (Lepikhin et al., 2020; Fedus et al., 2021; Yang et al., 2021; Lewis et al., 2021). With the growing interest of the Vision Transformer (Dosovitskiy et al., 2021; Liu et al., 2021; Wang et al., 2021; Xie et al., 2021b), MoE for vision (Riquelme et al., 2021; Wu et al., 2022) is also explored recently. However, there is still no self-supervised MoE model that can be trained on medium-sized datasets such as the ImageNet-1k.

Kudugunta et al. (2021); Ma et al. (2018) regard the MoE as a multi-task learning model, and use it for multi-language translation and recommendation systems, respectively. In this paper, we show that for self-supervised learning on images, an additional clustering component is crucial in the learning of a highly performant MoE model. Moreover, while the downstream tasks should follow the pre-training task in (Kudugunta et al., 2021; Ma et al., 2018), the MoCE can be used with any downstream task due to its unsupervised pre-training. Puigcerver et al. (2020) shares a similar setting with us, but their model is pre-trained in a supervised learning manner. Moreover, their mapping between experts and data is pre-defined and fixed during training, while that for the MoCE is learned dynamically and achieves better performance.

**Multi-Task Learning**    aims to learn a model that is appropriate for multiple tasks. Hard-parameter sharing, which uses a shared backbone with multi-heads for the different tasks, has been shown to be effective on time series, language and graph data (Liu et al., 2019; Hu et al., 2019; McDermott et al., 2021). Gao et al. (2021) claims that the network design may further benefit from the use of task relationships, and trains masks for different tasks. However, they require the task information be available during model training, which is not possible for downstream tasks in SSL pre-training.

## 3    PROPOSED METHOD

In this section, we first empirically demonstrate the negative transfer phenomenon in MAE (Section 3.1). We then discuss the limitations of adopting TokenMoE (Riquelme et al., 2021) with MAE (Section 3.2), and propose the Mixture of Cluster-conditional Experts (MoCE), a novel paradigm achieving customized pre-training for various downstream tasks (Section 3.3).

### 3.1    NEGATIVE TRANSFER IN MASKED AUTOENCODER

In this section, we evaluate the transfer performance of MAE models pre-trained with data of different semantics on various downstream tasks. As in (Huh et al., 2016; Liu et al., 2022), we first split the ImageNet data into two disjoint subsets, Split-A and Split-B, based on the labels' semantic dissimilarities in the WordNet tree (Miller, 1998). Split-A mainly contains inanimate objects (such as cars and airplanes), while Split-B primarily involves organisms (such as plants and animals). We then pre-train MAEs on Split-A, Split-B and the full ImageNet without data annotation, and evaluate the three resulting models on 11 downstream tasks. See more implementation details in Section 4.1.

As shown in Figure 1(a), the MAE pre-trained with Split-A performs best on *Aircraft*, *Cars*, *SUN397* and *DTD*, while the MAE pre-trained with Split-B performs best on *Flowers*, *Food*, *Pets*, *Cifar-10* and *Cifar-100*. Only two of the eleven tasks (*Caltech* and *VOC*) benefit from using the full

Figure 2: Model design comparison between (a) TokenMoE (Riquelme et al., 2021) and (b) MoCE. Both methods utilize the multi-expert architecture with the main difference about the input of the gating network. MoCE adopts the corresponding cluster embedding of the current token as in Eqn. 4, instead of the token embedding in Eqn. 3.2. Therefore, each expert can be trained by semantically similar images to alleviate the negative transfer phenomenon.

ImageNet data. This suggests that for tasks whose semantics are close to inanimate objects, adding pre-training data from Split-B is not useful, and vice versa for tasks whose semantics are close to organisms. To conclude, the introduction of semantically irrelevant pre-training data may impede transfer performance for downstream tasks. This negative transfer phenomenon motivates us to develop an efficient and automatic paradigm for task-customized pre-training.

## 3.2 Exploring TokenMoE to Masked AutoEncoder

**Overview of TokenMoE.** TokenMoE (Riquelme et al., 2021) is a successful customized supervised pre-training model built upon the ViT (Dosovitskiy et al., 2021), which mainly consists of transformer blocks with alternating multi-head self-attention (MSA) and multi-layer perceptron (MLP). Specifically, the TokenMoE converts several transformer blocks to Mixture of Expert (MoE) blocks by expanding the MLP layer $N$ times, each of them is considered as an *expert* (denoted as $E_i(\cdot)$, $i = 1, 2, \ldots, N$). Conditional computation on the $N$ experts is controlled by a *gate*, which is a linear layer whose input is the token embedding $\boldsymbol{x}$, and the output is the top-$K$ probabilities on the experts: $G(\boldsymbol{x}) = TopK(\sigma(\boldsymbol{W}_g\boldsymbol{x} + \epsilon))$, where $K$ is the number of experts to be activated, $\boldsymbol{W}_g$ is the gate parameter, $\sigma$ is the softmax function, and $\epsilon \sim \mathcal{N}(0, \frac{1}{N})$. $TopK(\cdot)$ returns the $K$ largest entries of $\sigma(\boldsymbol{W}_g\boldsymbol{x} + \epsilon)$ unchanged but set the others to zero. Thus, each token is routed to its corresponding experts. The final output is represented as

$$\mathbf{y} = \sum_{i=1}^{N}[G(\boldsymbol{x})]_i E_i(\boldsymbol{x}). \tag{1}$$

As in (Riquelme et al., 2021), importance loss and load loss are also used to enforce a balanced use of the experts. Unless otherwise specified, we set $K = 1$ and $N = 8$ in all our experiments.

**Limitation of TokenMoE.** As will be shown in the experimental results (Table 3), naively adopting TokenMoE to the MAE cannot improve performance, even with intense hyper-parameter tuning and data augmentations (e.g., Repeat Augment (Hoffer et al., 2020) and RandAugment (Cubuk et al., 2020) with larger magnitude). Figure 3(a) shows the routing heatmaps of the pre-trained TokenMoE model. As can be seen, the routing process has little correlation with the ImageNet labels. Moreover, expert 3 is selected most of the time (91.8% of the classes). This degenerates the multi-expert network into a single-expert network. As demonstrated in Figure 1(b), we speculate that this is due to the use of low-level pixel values (instead of semantic class labels in the original TokenMoE) as reconstruction targets. This is also observed in Li et al. (2022b).

## 3.3 Mixture of Cluster-conditional Experts

To address the limitations of TokenMoE, we propose the Mixture of Cluster-conditional Experts (MoCE), which trains each expert in a semantic-aware manner. The procedure consists of data clustering, architecture and gate design, and deployment.

**Data Clustering.** To train each expert semantically, a clustering procedure is first performed to simulate the label partitioning in Section 3.1. With a pre-trained MAE model, we collect all the image features $f_i$'s (normalized to unit length $\|f_i\| = 1$), and represent the feature matrix as $F = [f_1, f_2, \ldots, f_n] \in \mathbb{R}^{d \times n}$, where $n$ is the number of images and $d$ is the dimension of the feature. The learnable cluster centroids are represented as $C = [c_1, c_2, \ldots, c_m] \in \mathbb{R}^{d \times m}$, (with $\|c_i\| = 1$), where $m$ is the desired number of clusters. The assignment of feature to clusters is computed as $A = F^T C$. Following Asano et al. (2019), let $Q \in \mathbb{R}^{m \times n}$ be the posterior distribution of clustering, whose objective is

$$\max_{Q} Tr(Q^T A) + \epsilon H(Q) \quad s.t. \quad Q\mathbf{1}_n = \frac{1}{m}\mathbf{1}_m, \ Q^T\mathbf{1}_m = \frac{1}{n}\mathbf{1}_n, \tag{2}$$

where $\mathbf{1}_m$ is the $m$-dimensional vector of all ones, $H$ is the entropy function, and the constraints force the clustering results to be balanced. $Q$ and $C$ are optimized iteratively. For a given $C$, $Q$ is solved by the Sinkhorn-Knopp algorithm (Cuturi, 2013); while for a given $Q$, $C$ is obtained by minimizing the cross entropy between $Q$ and $A$ with SGD. We take the final $C$ and $Q$ as the cluster centroids and clustering assignments, respectively. The implementation details are in Appendix A.1.

**Architecture.** The whole network is trained on the full ImageNet data, with each expert trained by images from selected clusters decided by the MoCE gates' routing results. As on average each data cluster has only a fraction of $1/K$ of the original sample size, the training time of each expert is also $K$ times shorter than the other parameters with dense modeling (e.g., MSA parameters (Riquelme et al., 2021)), we further adopt a distillation loss $\mathcal{L}_{distill}$, which is defined as the $\ell_2$ distance between the features generated by the whole network and each expert. This loss function can be formulated as

$$\min_{\theta} \sum_{i=1}^{m} \mathcal{L}_{MAE}(D_i; \theta_i) + \mathcal{L}_{distill}, \tag{3}$$

where $D_i$ is the $i$th cluster, $\theta_i$ is the parameter used for training $D_i$, and $\mathcal{L}_{MAE}(D_i; \theta_i)$ is the reconstruction loss for masked image modeling. $\theta_i$ consists of several experts in different layers, as explained in the following.

**Gate Design.** As in the TokenMoE, we replace several MLP layers in the ViT with layers equipped with MoCE gates. In TokenMoE, routings of the tokens to experts are considered separately. In MoCE, we route tokens from images of the same cluster to the same expert. The MoCE gate output can thus be written as

$$G(\mathbf{x}) = TopK(\sigma(W_g \cdot C_{[\mathbf{x}]} + \epsilon)), \tag{4}$$

where $W_g$ is the gate parameter, and $C_{[\mathbf{x}]}$ is the embedding of the cluster that $\mathbf{x}$ belongs to. Empirically, we find that the confidence of $G(\mathbf{x})$ (the max entry) is low and consequently, the mapping between clusters and experts varies a lot during pre-training. Inspired by the importance and load losses (Riquelme et al., 2021), we add the following loss $\mathcal{L}_{imbalance}$ to enhance the confidence of the gates. Since it makes $G(\mathbf{x})$ shaper, we call it *imbalance* loss.

$$\mathcal{L}_{imbalance} = -\sum_{i=1}^{n} \left(\frac{std(G(\mathbf{x})_i)}{mean(G(\mathbf{x})_i)}\right)^2, \tag{5}$$

For practical implementation, the loss is calculated over the samples in a batch. The imbalance loss penalizes on the negative variance of the gate confidence.

**Deployment.** On deployment, customized experts are selected from MoCE, and fine-tuned for each downstream task. As shown in Section 3.1, we prefer to use the experts that is pre-trained from data whose semantics is closest to that of the downstream task. This can be obtained by re-using the data clustering module. Specifically, we feed images for the downstream task through the pre-trained MAE model and collect all the image features as $F_{task}$. The assignment of downstream images to the clusters is then computed as $A_{task} = F_{task}^T C$. We select the largest cluster with assigned downstream images, and use the corresponding experts (a **sub-model** of the whole MoCE model) for deployment. In the case when only one expert is activated at each MoCE layer ($K = 1$), a regular ViT model is needed for downstream fine-tuning, which is much more efficient than MoE.

Table 1: Transfer accuracy (%) of self-supervised learning models on 11 downstream tasks.

| | Aircraft | Caltech | Cars | C10 | C100 | DTD | Flowers | Food | Pets | SUN | VOC | Avg. |
|---|---|---|---|---|---|---|---|---|---|---|---|---|
| *ResNet-50* | | | | | | | | | | | | |
| BYOL | 82.39 | 90.12 | 87.33 | 96.28 | 82.15 | 74.57 | 95.96 | 82.13 | 88.52 | 64.41 | 83.97 | 84.35 |
| DeepCluster-v2 | 78.75 | 90.51 | 86.33 | 96.48 | 82.28 | 75.43 | 96.16 | 83.68 | 90.33 | 66.68 | 81.37 | 84.36 |
| *Vision Transformer* | | | | | | | | | | | | |
| Supervised | 76.55 | 89.98 | 86.19 | 96.79 | 83.96 | 75.09 | 93.94 | 85.17 | 92.54 | 64.54 | 87.22 | 84.72 |
| DINO | 66.50 | 91.65 | 76.37 | 98.12 | 86.69 | 75.73 | 96.40 | 93.77 | 93.97 | 59.33 | 86.62 | 84.10 |
| MoCo v3 | 76.29 | 91.64 | 85.18 | 97.99 | 86.98 | 72.64 | 95.33 | 83.94 | 92.35 | 65.54 | 84.21 | 84.74 |
| BEiT | 53.16 | 79.02 | 68.11 | 94.34 | 73.54 | 68.04 | 91.33 | 79.59 | 84.02 | 56.13 | 65.65 | 73.90 |
| MAE | 72.38 | 90.47 | 83.51 | 95.69 | 68.40 | 75.48 | 96.10 | 79.98 | 92.35 | 62.43 | 84.79 | 81.96 |
| MAE* | 72.71 | 91.24 | 84.47 | 96.15 | 77.33 | 75.05 | 96.25 | 80.49 | 92.78 | 62.46 | 85.02 | 83.09 |
| MoCE (Ours) | 78.73 | 90.61 | 88.56 | 97.79 | 84.68 | 74.04 | 96.94 | 86.24 | 93.07 | 65.05 | 85.26 | $\mathbf{85.54}^{+2.45}$ |

Table 2: Transfer accuracy (%) on detection and segmentation.

| Method | ADE20K | COCO | | | | | |
|---|---|---|---|---|---|---|---|
| | mIoU | $AP^{bb}$ | $AP^{bb}_{50}$ | $AP^{bb}_{75}$ | $AP^{mk}$ | $AP^{mk}_{50}$ | $AP^{mk}_{75}$ |
| Supervised | 46.9 | 48.8 | 68.7 | 52.7 | 42.5 | 65.9 | 45.5 |
| DINO | 46.9 | 49.5 | 69.1 | 53.6 | 42.9 | 66.0 | 46.3 |
| MoCo v3 | 46.8 | 47.2 | 66.9 | 50.8 | 41.1 | 63.6 | 44.1 |
| BEiT | 45.6 | 40.8 | 59.4 | 44.1 | 36.0 | 56.8 | 38.2 |
| MAE | 48.1 | 50.6 | 69.4 | 55.0 | 43.8 | 66.6 | 47.5 |
| MoCE | **48.3** | **51.1** | **69.8** | **55.4** | **44.2** | **67.0** | **48.1** |

# 4 EXPERIMENTS

In this section, we first introduce the setup of pre-training and fine-tuning stage of MoCE in Sec. 4.1. Then we demonstrate the effectiveness of MoCE by evaluating the pre-trained models on a collection of 11 downstream tasks with detailed analysis of our MoCE superior to vanilla MAE and TokenMoE in Sec. 4.2. Finally we take ablation studies on the key components of MoCE in Sec. 4.3.

## 4.1 SETUP

For all experiments, we replace two MLP layers with MoCE layers in the original ViT-B (Dosovitskiy et al., 2021). Following Wu et al. (2022), layers with the greatest gradient magnitude are selected (which are the last two MLP layers in our experiments). Unless otherwise specified, the number of experts is 8 and the number of clusters is 256. Our model utilizes the officially released 1600-epoch pre-trained MAE model[2] and continues to train for an extra 200 epochs. Each expert is initialized by the corresponding dense model with a small weight perturbation. The training procedure mainly follows that of MAE, except that we multiply the base learning rate by 0.1. All regularization loss weight is set to 0.01 by default.

To ensure a fair comparison with the vision transformer on downstream classification tasks, we mainly follow the hyper-parameter settings in (Dosovitskiy et al., 2021; Riquelme et al., 2021) and the benchmark settings in (Ericsson et al., 2021). The proposed model is compared with various self-supervised models, including DINO (Caron et al., 2021), MoCo v3 (Chen et al., 2021b), BEiT (Bao et al., 2022), and the highly-performant ResNet-50 models of BYOL (Grill et al., 2020) and DeepCluster-v2 (Caron et al., 2018). We also compare with the supervised pre-trained model DeiT (Touvron et al., 2021). To make a fair comparison of training time, we continue to train a 1600-epoch pre-trained MAE for 200 epochs with total ImageNet as our baseline, and is denoted as MAE* in Table 1. For detection and segmentation tasks, following Bao et al. (2022), we perform experiments on ADE20K (Zhou et al., 2019) and COCO (Lin et al., 2014). We utilize the officially released checkpoints for all baselines. Details are in Appendix A.5.

---

[2]https://github.com/facebookresearch/mae

Table 3: Transfer accuracy of MAE, TokenMoE, SDR and MoCE. SDR(ViT) is our re-implementation of SDR under ViT. We observe that TokenMoE cannot outperform vanilla MAE, while SDR(ViT) achieves better performance, which is further outperformed by MoCE.

| | Aircraft | Caltech | Cars | C10 | C100 | DTD | Flowers | Food | Pets | SUN | VOC | Avg. |
|---|---|---|---|---|---|---|---|---|---|---|---|---|
| MAE* | 72.71 | 91.24 | 84.47 | 96.15 | 77.33 | 75.05 | 96.25 | 80.49 | 92.78 | 62.46 | 85.02 | 83.09 |
| TokenMoE | 70.51 | 89.70 | 81.40 | 95.18 | 76.44 | 73.67 | 95.09 | 77.45 | 90.71 | 61.12 | 80.15 | 81.04 |
| SDR | 75.77 | 89.73 | 86.65 | 95.31 | 83.60 | 73.62 | 95.53 | 84.77 | 91.25 | 64.64 | 83.51 | 84.03 |
| SDR(ViT) | 76.57 | 90.04 | 86.95 | 96.92 | 81.42 | 73.09 | 96.14 | 82.90 | 92.65 | 64.40 | 85.37 | 84.22 |
| MoCE | 78.73 | 90.61 | 88.56 | 97.79 | 84.68 | 74.04 | 96.94 | 86.24 | 93.07 | 65.05 | 85.26 | **85.54** |

## 4.2 RESULTS

**Transfer Results.** The classification transfer performance of various self-supervised models are shown in Table 1. As can be seen, MoCE achieves a 2.45% improvement over MAE* and reaches the state-of-the-art averaged accuracy, demonstrating the effectiveness of the task-customized pre-training paradigm. On fine-grained datasets such as *Aircraft*, *Cars* and *Food*, MoCE outperforms the baseline model by a large margin. This is because these fine-grained tasks are similar to only a subset of the pre-training dataset. Hence, MoCE can alleviate negative transfer by using the model that is trained by the cluster most similar to the particular downstream task. On the other hand, MoCE shows only limited improvement on tasks such as *Caltech*, *Cifar-10* and *VOC*. These tasks are more general and contain images covering the various semantics in the pre-training dataset, and thus negative transfer does not exist.

Table 2 shows the transfer performance on the detection and segmentation tasks. As can be seen, MoCE outperforms MAE and the other baselines (including the supervised one), and achieves state-of-the-art results.

**Comparison between MoCE, TokenMoE, MAE and SDR.** In this experiment, we compare MoCE with the following models: (i) MAE, (ii) TokenMoE, (iii) SDR (Liu et al., 2022), a task-customized model that aims at alleviating negative transfer, and (iv) SDR(ViT), which re-implements SDR with the ViT architecture. Table 3 shows the transfer accuracy on 11 downstream tasks. As can be seen, TokenMoE performs even worse than MAE, suggesting that naively adopting MoE to MAE is not desirable. Both MoCE and SDR(ViT) outperform MAE, demonstrating the effectiveness of task-customized methods for alleviating negative transfer. MoCE further outperforms SDR(ViT), indicating the importance of self-adaptive routing.

Figure 3(d) shows the peak signal-to-noise ratio (PSNR) (Sara et al., 2019), which reflects the generation quality of these autoencoder models. MoCE exhibits improvement over TokenMoE and MAE on most datasets. We also provide the comparisons in the case of a fair parameter count, large architectures, and training from scratch in the Appendix A.2, A.3 and A.4, respectively.

**Analysis on experts.** Figure 3(a) and Figure 3(c) show the routing heatmaps for TokenMoE and MoCE, respectively. As can be seen, routing of the TokenMoE experts has little correlation with semantics. On the other hand, each MoCE expert is trained by several classes, showing a more balanced assignment of images to experts. This verifies that the improvement of MoCE is due to more effective learning of the experts. Moreover, notice that the importance loss and load balance loss (Riquelme et al., 2021) are applied and indeed work as "expected" because they are only applied with respect to patch tokens instead of semantic classes. On the other hand, MoCE can balance the experts both at the token level and semantic level.

Figure 3(b) shows example pre-training samples for 3 random MoCE experts. Note that expert 1 is mostly trained by images containing clothes, experts 2 is pre-trained mostly by bird images, while expert 3 is pre-trained mostly by dog images.

Next, we show that each expert is trained by samples with similar semantics. Following (Mikolov et al., 2013), we select the label set used by each expert, and then compute the $\ell_2$ distances between CLIP embeddings (Radford et al., 2021a) of labels used by the same expert and by different experts. The average distance between labels used by the same expert is 0.84, while that between labels

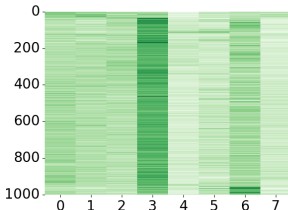

(a) Routing heatmap for TokenMoE experts.

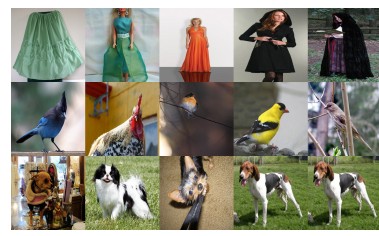

(b) Examples pre-training samples for expert 1 (top), expert 2 (middle), and expert 3 (bottom).

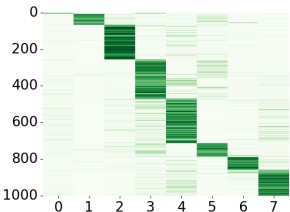

(c) Routing heatmap for MoCE experts.

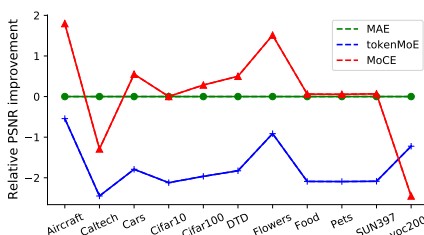

(d) Relative PSNR improvement over MAE.

Figure 3: (a),(c): Routing heatmaps for experts in TokenMoE and MoCE. The x-axis is the expert ID, and the y-axis is the ImageNet semantic label ID. Darker green means a higher proportion of tokens belonging to the corresponding class are allocated to the expert. The label is sorted differently in each figure to make it readable. (b): Example samples from the pre-training dataset of 3 MoCE experts. (d): Relative PSNR improvement of TokenMoE and MoCE over MAE for each downstream task.

used by different experts is 0.92, indicating that the MoCE gate automatically aggregates labels with similar semantics to each expert, thus benefiting downstream transfer.

**Training and testing efficiency.** Table 4 compares the efficiencies of MAE, TokenMoE and MoCE during training and testing. As can be seen, all of them have similar FLOPs. However, TokenMoE needs to use the whole model during both training and testing, while MoCE only needs to use a single expert and thus halves the required number of parameters when testing. In addition, the training and testing speeds are improved by respectively 18% and 37%, which is attributed to the reduction of token shuffle operations as tokens in the same image do not need to be split and are dispatched to the same expert, significantly reducing the communication overhead.

## 4.3 ABLATION

**Search method.** When a downstream task arrives, it is expensive to fine-tune all experts to choose the best one. To find the task-customized expert ($K = 1$), we compare the method proposed in Section 3.3 with (i) early stop, (ii) KNN (Liu et al., 2022), (iii) LogME (You et al., 2021). The experiment is performed on the task with the most samples (*Food*), the task with the least samples (*Flowers*), and the one with a medium number of samples (*Aircraft*). For comparison, we additionally show the performance of the best and worst experts based on an exhaustive search. As can be seen in Table 5, MoCE performs stably among different sizes of the dataset, and the search cost is negligible as we only need to infer the downstream task once and feed it to the clustering module. This illustrates another advantage of combining clustering and pre-training in a single paradigm.

**MoCE Architectures.** In this experiment, we study the different architecture hyper-parameters of MoCE in three aspects. First, we vary the number of experts in each MoCE layer. As can be seen from Table 6, using more experts leads to consistent improvement on the accuracy.

Next, we vary the location of the MoCE layers. As mentioned in Section 4.1, we select the MoCE layers based on the gradient magnitudes. In the experiments, MoCE selects the 11th and 12th MLP layers. On the other hand, TokenMoE chooses the last 2 even-numbered (i.e., 10th and 12th) MLP

Table 4: Efficiency during training (top) and testing (bottom).

| | MAE | TokenMoE | MoCE |
|---|---|---|---|
| Params (M) | 111.91 | 178.03 | 178.03 |
| FLOPs (G) | 9.80 | 9.81 | 9.81 |
| Speed↑ | 1.41x | 1x | **1.18x** |
| # Params (M) | 85.88 | 152.00 | 85.88 |
| FLOPs (G) | 16.88 | 16.88 | 16.88 |
| Speed↑ | 1.37x | 1x | **1.37x** |

Table 5: The search cost (in GPU hours) for different expert search algorithms.

| | Aircraft | Flowers | Food | GPU hours |
|---|---|---|---|---|
| Best | 79.92 | 97.96 | 86.24 | 288 |
| Worst | 69.84 | 94.97 | 81.51 | 288 |
| Early stop | 77.00 | 96.83 | 85.33 | 144 |
| KNN | 71.40 | 95.10 | 83.32 | 5 |
| LogME | 73.84 | 96.54 | 85.11 | 5 |
| MoCE | **78.73** | **96.94** | **86.24** | **1** |

Table 6: Accuracies with different numbers of experts in a MoCE layer. (default setting used is in bold).

| # experts | Acc (%) |
|---|---|
| 1 | 83.09 |
| 2 | 83.01 |
| 4 | 84.22 |
| **8** | **85.54** |

Table 7: Accuracies with different numbers and locations of the MoCE layers (default setting used is in bold).

| # MoCE layers | Acc (%) |
|---|---|
| 1 | 83.09 |
| 2 (10th & 12th) | 83.08 |
| **2 (11th & 12th)** | **85.54** |
| 4 | 85.59 |

Table 8: Accuracies with different numbers of clusters (default setting used is in bold).

| # clusters | Acc (%) |
|---|---|
| 16 | 82.00 |
| 64 | 84.02 |
| **256** | **85.54** |
| 512 | 85.33 |

layers. Furthermore, we exhibit the performance with only 1 and 4 MoCE layers, which are also selected based on the gradient magnitudes. As shown in Table 7, we notice that it is essential to choose the right layer to be MoCE layer. Adding more MoCE layers shows little improvement.

We also train MoCE with different numbers of clusters. As shown in Table 8, the accuracy increases up to 256 clusters, and then begins to drop. We hypothesize that with a moderate number of clusters, MoCE can produce a variety of task-customized models. With even more clusters, the number of experts become the bottleneck and performance starts to saturate.

## 5 Conclusion

In this work, we first show that the negative transfer phenomenon exists in the prevailing self-supervised learning method MAE through extensive experiments. It will impede the scalability of MAE as more pre-training data may instead degenerate the downstream performance. In order to tackle the problem, we introduce Mixture of Expert to MAE as the multi-experts design can equip MAE with different ability that aids transfer. However, different from supervised pre-training, To-kenMoE suffers from the fact that the gate shows no correlation to the semantics and the transfer ability is not improved. Based on this, we propose MoCE to explicitly train each expert with different clusters through the MoCE gate design and several losses to stabilize the training process. A search algorithm for selecting the best model for transfer is also proposed based on the clustering priors. Extensive experiments show that MoCE trains each expert with meaningful semantics and achieves state-of-the-art transfer performance on a collection of 11 downstream tasks and both detection and segmentation tasks. It is the first work that successfully trains a self-supervised learning MoE model on ImageNet only. We hope such a design will motivate more research on the self-supervised MoE models.

### Acknowledgments

We gratefully acknowledge the support of MindSpore, CANN (Compute Architecture for Neural Networks) and Ascend AI Processor used for this research.

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

# A APPENDIX

## A.1 DETAILS FOR CLUSTERING.

For data clustering, the features are computed by inferring the pre-train MAE, and the matrix $\mathbf{Q}$ and $\mathbf{C}$ are solved by the Sinkhorn-Knopp algorithm and SGD optimizer iteratively. For the Sinkhorn-Knopp algorithm, we set the iteration number as 3. The learning rate of SGD is set to 0.1, the momentum is 0.9 and weight decay is set to 0.9 for the sparse assignment of cluster results. We train 10 epochs in total and it costs 3 minutes and 20 seconds on average for a single GPU.

## A.2 COMPARISON UNDER FAIR PARAMETER COUNTS.

The setting used in our work focuses on a fair comparison of **FLOPs**, referring to Table 4 in the main paper. Since TokenMoE and MoCE always activate only one expert throughout the whole pre-training and fine-tuning procedure, the FLOPs value is maintained close to MAE. Apart from this criterion, we further provide the comparison on equal **parameter** counts. As shown in Table 9, we train MAE under the same parameter count as the whole model of MoCE, and MoCE still outperforms MAE consistently.

Table 9: Comparison of MAE and MoCE under equal parameter counts. We train MAE with a larger model that shares the same parameter count as the whole model of MoCE.

|      | # Params | Aircraft | Caltech | Cars | C10 | C100 | DTD | Flowers | Food | Pets | SUN | VOC | Avg. |
|------|----------|----------|---------|------|-----|------|-----|---------|------|------|-----|-----|------|
| MAE  | 178.03   | 74.43    | 90.30   | 85.50 | 96.90 | 83.80 | 74.84 | 96.30 | 81.86 | 92.97 | 62.98 | 85.51 | 84.13 |
| MoCE | 178.03   | 78.73    | 90.61   | 88.56 | 97.79 | 84.68 | 74.04 | 96.94 | 86.24 | 93.07 | 65.05 | 85.26 | **85.54** |

## A.3 MOCE FOR LARGER ARCHITECTURE.

Here we provide the analysis on the larger architecture(ViT-L, 2.57× larger than ViT-B) to explore the scalability of MoCE. We first demonstrate that negative transfer still exists for larger architecture by training MAE with total ImageNet, Split-A and Split-B following the same setting in Sec. 3.1. As shown in the first three rows of Table. 10, a similar phenomenon is observed that MAE-L trained by Split-A performs better in *Aircraft*, *Cars*, *DTD* and *SUN* while Split-B in *Flowers*, *Food*, and *Pets*. On the other hand, MoCE-L can still alleviate the problem and therefore transfers better. We believe that the negative transfer phenomenon mainly exists when a common pre-trained model is used for various downstream tasks, due to the inevitable semantic gaps between the pre-training and downstream datasets, rather than the architecture.

Table 10: Comparison of MAE and MoCE on ViT-L. We also train MAE with 2 subsets of ImageNet, namely Split-A and Split-B, following the same setting mentioned in Sec. 3.1. This table shows that negative transfer still exists on larger architectures, while MoCE can alleviate this problem and achieve better transfer results.

|                   | Aircraft | Caltech | Cars | C10 | C100 | DTD | Flowers | Food | Pets | SUN | VOC | Avg. |
|-------------------|----------|---------|------|-----|------|-----|---------|------|------|-----|-----|------|
| MAE-L (full set)  | 74.30    | 93.97   | 88.60 | 97.85 | 82.47 | 77.61 | 96.67 | 81.22 | 93.97 | 67.99 | 88.30 | 85.72 |
| MAE-L (Split-A)   | 79.70    | 91.59   | 89.33 | 96.97 | 80.38 | 78.67 | 95.44 | 82.97 | 92.49 | 68.73 | 82.41 | 85.33 |
| MAE-L (Split-B)   | 73.42    | 90.80   | 86.00 | 96.18 | 78.73 | 77.34 | 96.75 | 83.63 | 94.92 | 66.06 | 85.85 | 84.52 |
| MoCE-L            | 87.04    | 94.86   | 90.72 | 98.29 | 87.49 | 76.65 | 97.38 | 88.21 | 95.89 | 69.49 | 89.13 | $\mathbf{88.65}^{+2.93}$ |

## A.4 PERFORMANCE OF MOCE WITHOUT PRE-TRAINING.

We provide results of MoCE trained from scratch for 200 epochs and 1600 epochs in Table 11. In this experiment, for clustering, we first pre-train MAE for 50 epochs and perform clustering. We then train MoCE from scratch for 200 epochs and 1600 epochs based on the clustering results. Although it is a common practice to utilize pre-trained dense models as initialization to accelerate pre-training (Wu et al., 2022; Bai et al., 2022), MoCE still outperforms MAE consistently in various downstream tasks when trained from scratch.

Table 11: Comparison of MAE and MoCE both training from scratch for 200 epochs (first two rows) and 1600 epochs (last two rows).

|      | Aircraft | Caltech | Cars | C10 | C100 | DTD | Flowers | Food | Pets | SUN | VOC | Avg. |
|------|----------|---------|------|-----|------|-----|---------|------|------|-----|-----|------|
| MAE  | 64.73 | 85.91 | 77.10 | 92.92 | 72.50 | 73.30 | 93.11 | 73.14 | 88.70 | 57.84 | 73.27 | 77.50 |
| MoCE | 71.16 | 90.55 | 82.46 | 96.06 | 76.56 | 74.57 | 95.70 | 79.67 | 92.58 | 62.20 | 84.25 | **82.34** |
| MAE  | 72.38 | 90.47 | 83.51 | 95.69 | 68.40 | 75.48 | 96.10 | 79.98 | 92.35 | 62.43 | 84.79 | 81.96 |
| MoCE | 78.75 | 91.64 | 87.04 | 97.15 | 83.12 | 73.62 | 96.08 | 83.84 | 93.06 | 65.49 | 85.81 | **85.05** |

## A.5 EVALUATION DETAILS FOR DOWNSTREAM TASKS.

**Classification.** We mainly follow the settings of Ericsson et al. (2021).to make a fair comparison. Specifically, all models are trained by SGD with a momentum of 0.9. Weight decay is set to be 0 and the learning rate is searched among [1e-4, 3e-4, 1e-3, 3e-3, 1e-2, 3e-2, 1e-1, 3e-1]. Each model is fine-tuned for 2500 steps with cosine learning rate decay, a batch size of 64, and $224 \times 224$ resolution. We fine-tune each model 3 times and report the average performance. We find such a setting generates a stable result.

**Semantic segmentation.** We evaluate MoCE on the semantic segmentation task that aims to predict the class for each pixel in the image. We report the metric of mean Intersection of Union (mIoU) averaged over all semantic categories in ADE20K (Zhou et al., 2019). We choose the best expert by applying ADE20K images to our clustering module and selecting the cluster that contains the most images. We use Adam (Loshchilov & Hutter, 2017) as the optimizer. The learning rate is set to 1e-3 with layer-wise learning rate decay (Clark et al., 2020) to be 0.65. We conduct fine-tuning for 160K steps. The batch size is 16. The detailed hyper-parameters can refer to Bao et al. (2022).

**Detection and Instance segmentation** are also evaluated on COCO (Lin et al., 2014). We follow the same deployment method as the one used in the semantic segmentation task to choose the best expert. Following iBOT (Zhou et al., 2021) we adopt the Cascade Mask R-CNN (Cai & Vasconcelos, 2018; He et al., 2017) and the multi-scale training. The shorter side is randomly resized between 480 and 800 while the longer one is no longer than 1333. The batch size is 16, and the initial learning rate is 1e-4. The layer-wise learning rate decay ratio (Clark et al., 2020) is set to 0.75. We train the model for 12 epochs and decrease the learning rate by 10x at epoch 9 and 11.

