# OpenReview forum: "Task-customized Masked Autoencoder via Mixture of Cluster-conditional Experts"
_ICLR.cc/2023/Conference — ICLR 2023 notable top 25%_

### Official Review · Reviewer_Emir · 2022-10-25

**Confidence:** 3
**Correctness:** 3
**Technical Novelty And Significance:** 3
**Empirical Novelty And Significance:** 3
**Recommendation:** 6

**Clarity, Quality, Novelty And Reproducibility:**

The paper is written clearly and easy to follow.

It’s somewhat surprising to see TokenMoE is much lower than MAE (Table 2). Do you have any intuition as to why? Taking a look at, e.g. Fig 3a, something might happen with TokenMoE, leading to imbalancing.

Any reason why not using, say, t-SNE to visualize the clusters?

No code or appendix is provided so it’s hard to judge the reproducibility.


**Strength And Weaknesses:**

Strength: 1/ Having a good motivation on an important application, where we should take advantage of pretrained models to save cost.  2/ The testing efficiency is really good.

Some Weaknesses

1/ About cost saving: the proposed solution has a non-negligible overhead of data clustering, which is itself an optimization problem. My understanding is this overhead is not included in the training analysis (please correct if I am wrong).

2/ It’s not entirely clear about the location and number of MoCE layers chosen even with Fig 4b and Table 6,  e.g. the intuition as to why placing them on, say, 10th and 12th layers are good but not before that? This is also along with the claim of hierarchical learning by the model concerning Fig 4b. Why don’t we try a couple of other patterns such as interleaving odd and even layers, or early in the layers while the input is already arranged in clusters, which is somewhat more intuitive?


**Summary Of The Paper:**


The paper has a good motivation of transferring a pretrained model into diverse tasks by tackling the problem of negative transfer. The solution is having an abstraction layer of semantic, where input is grouped into different clusters based on their semantics before routing to experts. Experiments show a very good improvement of 2.54% on ImageNet.


**Summary Of The Review:**


The paper motivates a good problem in transferring a vision task and proposes a good solution in which semantics is used to train the mixture-of-expert model, and perform well with diverse tasks. Although there are some parts that need more clarification, generally it’s a good and well written paper.

---

> ### Author Response · Authors · 2022-11-14
> **Response to Reviewer Emir**
>
> Thank you for your valuable and constructive reviews. Here we address your questions one by one.
>
>
>
> **Q.1. The cost analysis for data clustering.**
>
> Thank you for pointing this out.
>
> 1. The computation cost of the offline clustering has already been included in Table 3 (Row ‘Speed’) of the main paper.
> 2. Specifically, the clustering procedure requires only 3min20s on a single GPU, which is negligible compared to the pre-training procedure of MoCE of 10 GPU days.
>
> 3. We have updated the details for clustering in Appendix A.1
>
>
>
> **Q.2. Analysis of the location and number choice for MoCE layers**.
>
> R.2.
>
> 1. In Table 6 in the main paper, we have provided an ablation to study the effect of different numbers and positions of MoCE layers.
>
> 2) Since it is hard for us to grid search the position factor, we mainly follow the setting of [1] and [2] to set the last-2 even layers(10-th and 12-th) and the layer with the greatest gradient magnitude (11-th and 12-th, default for MoCE) respectively.
> 3) Moreover, as observed in [1], gate selection in early layers close to the input tends to be random, which is why we mainly focus on replacing deep ViT layers with MoCE ones.
> 4) As suggested, we also provide the transfer results of applying MoCE to the 1-st and 2-nd layers and observe a performance drop, as shown in the table below.
>
>
>
> Table 1. Average performance over 11 downstream tasks when applying MoCE to different layers.
>
> | Position      | Acc.  |
> | ------------- | ----- |
> | 1-st & 2-nd   | 82.85 |
> | 10-th & 12-th | 83.08 |
> | 11-th & 12-th | 85.54 |
>
>
>
> **Q3. It’s somewhat surprising to see TokenMoE is much lower than MAE (Table 2). Do you have any intuition as to why? Taking a look at, e.g. Fig 3a, something might happen with TokenMoE, leading to imbalancing.**
>
> We speculate on the cause of the unsatisfactory performance of TokenMoE coming from the usage of data augmentation. As demonstrated in [1], TokenMoE needs strong data augmentation on the dataset like ImageNet, while stronger augmentation may degenerate MAE’s performance as stated in [3]. Such conflict may cause a worse performance of TokenMoE than MAE, which makes it not trivial to combine both.
>
> On the other hand, we observe that TokenMoE assigns patches to experts mainly based on the low-level features instead of semantics, as illustrated in Fig.1(b), which cannot generate a balanced assignment with respect to semantic classes even with the load balance loss, as shown in Fig. 3(a). Thus, TokenMoE can not successfully deal with the negative transfer problem, motivating us to propose MoCE that experts are trained by distinct semantics.
>
>
>
> **Q4. Why not use t-sne to visualize clusters?**
>
> We have already provided the visualization results of the learnt cluster assignments. Please refer to Fig.4(b) in the main paper for more details.
>
> [1] Riquelme, Carlos, et al. "Scaling vision with sparse mixture of experts." NIPS 2021.
>
> [2] Wu, Lemeng, et al. "Residual Mixture of Experts." Arxiv 2022.
>
> [3] He, Kaiming, et al. "Mask Autoencoder are Scaleable Vision Learners." Arxiv 2021.

---

> > ### Comment · Reviewer_Emir · 2022-11-20
> > **Thank you!**
> >
> > I personally thank the authors for addressing my questions and suggestions, and that you did an extended job in responding to other 3 reviewers and revised the paper due to their recommendations as well. It's a good paper and I am personally still supporting it to appear at the conference.

---

### Official Review · Reviewer_Wv7b · 2022-10-25

**Confidence:** 3
**Correctness:** 3
**Technical Novelty And Significance:** 2
**Empirical Novelty And Significance:** 2
**Recommendation:** 6

**Clarity, Quality, Novelty And Reproducibility:**

This paper is based on two previous works, i.e. (Liu et al., 2022) and TokenMoE. It offers some insights into the Mixture of Experts on MAE pre-training. The overall technical novelty is kind of incremental.

There is no code attached to this paper, but it will be possible to reproduce the results based on the descriptions and experimental details.




**Strength And Weaknesses:**

# Strength
- This paper proposed practical techniques to address the expert collapse issue in TokenMoE.
- The proposed MoCE improves the baseline performance on 11 downstream tasks, showing the effectiveness of the proposed method.
- Reasonable visualization of the expert routing and expert visualization is shown in the experiment.
- Several ablation studies are conducted in Table 4-7.


# Weaknesses
- Although better performance is achieved on the benchmark dataset, the technical contribution is kind of incremental. The main algorithm re-uses multiple design components of (Liu et al., 2022) and applies them to a slightly different scenario, i.e. MAE self-supervised training with a Mixture of Experts. Compared with (Liu et al., 2022), the improvements are (1) exploring the new self-supervised paradigm MAE, (2) applying to the Transformer backbone, and (3) different routing mechanisms with MoE. Both (1) and (2) are straightforward extensions to new technologies. As to (3), it offers some insights and benefits on self-supervised pre-training.
- As mentioned above, the proposed method heavily relies on design modules from (Liu et al., 2022), e.g. Data Clustering (Eq. (3)), Distillation term. However, most of the methodology and experimental comparison are conducted with respect to TokenMoE, a specific supervised MoE model. As also verified by the authors in Figure 3(a), TokenMoE is not specifically designed for the setting of self-supervised pre-training. Therefore, more comparisons and discussions should be made w.r.t. (Liu et al., 2022) rather than TokenMoE.
- TokenMoE used a load loss to balance the expert assignment, so it looks a little strange in Figure 3(a) the expert collapsing.
- In Table 3, the results for MAE are missing.
- It seems the results of DeepClusterV2 and BYOL in Table 1 are different from the results from (Liu et al., 2022), both on Resnet 50.
- Other minors:
> "decades(Jacobs et al.," and "processing(Lepikhin et al., 2020", missing space before left parentheses.
> Equ. 5 is not so common, usually Eqn. 5 or Eq. 5.
> In Eq. (3), $Tr(Q^TA), Q\in \mathbb{R}^{n\times m}, A\in \mathbb{R}^{n\times m}$, please check if this is correct. A more common notation for Sinkhorn is using matrix product $<Q, A>$.


**Summary Of The Paper:**

This paper investigates practical techniques to apply the Mixture of Experts (MoE) model to the MAE-based pre-training paradigm.

Taking a supervised MoE model for ViT as a baseline, the authors introduced data clustering as a strong baseline for experts' partition and assignment.

Several additional terms, such as imbalance regularization and distill loss, are also proposed in the self-supervised MoE design.

On 11 downstream tasks, the proposed method outperforms MAE, supervised TokenMoE and several recent self-supervised single model methods (non-MoE).


**Summary Of The Review:**

Although this paper obtains improved results over MAE and other self-supervised methods, the technical contribution is kind of incremental w.r.t (Liu et al. 2022) and TokenMoE.
And there is a lack of discussion and comparison w.r.t (Liu et al. 2022)

Overall, this paper is below the acceptance threshold.

------------------------------
After Rebuttal:
The authors addressed most of my concerns and made significant efforts. I would like to see this paper accepted and increased the overall score.

---

> ### Author Response · Authors · 2022-11-14
> **Response to Reviewer Wv7b**
>
> Thank you for your valuable and constructive reviews. Here we address your questions one by one.
>
> **Q.1 & Q.2. Comparison to Liu et al., (2022).**
>
> R.1 & R.2. We clarify that the main differences between our method and Liu et al., (2022) include:
>
> 1. Liu et al., (2022) fixes the assignment between the sub-networks and data clusters during model training. In contrast, MoCE dynamically maps the assignment between sub-networks and clusters, which increases the diversity of the sub-networks and leads to better performance on downstream tasks.
> 2. Liu et at., (2022) requires multi-stage data-aware progressive training. On the other hand, MoCE achieves efficient end-to-end training that greatly reduces the training difficulty.
> 3. After training, Liu et at., (2022) searches for the best sub-network brute-forcely through the KNN classifier. On the other hand, MoCE utilizes the clustering module during the deployment stage for efficient semantic-aware testing, which is a special design for MoCE.
> 4. Additional regularization terms, including distillation loss and imbalance loss, are proposed to stabilize and improve the training stability with dynamical routing.
>
> Note that the above clarifications have also been included in the ‘RELATED WORK’ in Sec. 2 of the main text.
>
> For quantitative comparison,
>
> 1. We provide the performance of the original SDR model in Table 2 of the revised paper.
>
> 2. We further implement their idea in transformer architecture, notated as SDR (ViT), and report it’s performance in Table 2 of the revised paper. Observation can be made that MoCE obtains consistent improvement over SDR.
>
> 3. We have also analyzed the deployment efficiency between SDR and MoCE in Table.4, where MoCE achieves better deployment efficiency than the SDR using KNN search.
>
>
>
> **Q.3. TokenMoE used a load loss to balance the expert assignment, so it looks a little strange in Figure 3(a) the expert collapsing.**
>
> R.3. The load balance loss is indeed working as “expected”, since it is applied with respect to patch tokens instead of semantic classes.
>
> Specifically, the proportional number(%) of patch tokens processed by 8 experts are 13.2, 12.3, 12.3, 18.7, 9.9, 10.3, 13.0, and 10.2, which is balanced. In contrast, Figure 3(a) of the main paper compares the semantic class assignment of each expert (i.e., the y-axis of Figure 3(a) is the semantic class ID), which suggests that expert 3 covers most of semantics while others cover little, thus the experts are collapsed. On the other hand, MoCE can balance the experts both at the token-level and semantic-level, as demonstrated in Figure 3(b).
>
>
>
> **Q.4. In Table 3, the results for MAE are missing.**
>
> R.4. Thanks for your kind reminder and we have provided the results of MAE in Table 3 in the revised version. The results are also provided as follows.
>
>
>
> ▾ Table 1. Training (top 3 rows) and testing (last 3 rows) efficiency comparison.
>
> |           | MAE    | TokenMoE | MoCE   |
> | --------- | ------ | -------- | ------ |
> | Params(M) | 111.91 | 178.03   | 178.03 |
> | FLOPs(G)  | 9.80   | 9.81     | 9.81   |
> | Speed ↑   | 1.41x  | 1x       | 1.18x  |
> | Params(M) | 85.88  | 152.00   | 85.88  |
> | FLOPs(G)  | 16.88  | 16.88    | 16.88  |
> | Speed ↑   | 1.37x  | 1x       | 1.37x  |
>
>
>
> **Q.5. It seems the results of DeepClusterV2 and BYOL in Table 1 are different from the results from (Liu et al., 2022), both on Resnet 50.**
>
> R.5. The results are different from (Liu et al., 2022) since we utilize a different transfer setting. Following MAE, we utilize the fully fine-tuning by default in this paper, while SDR conducts experiments using the linear protocol, which only tunes the linear classification head with the parameters of the backbone network fixed. Compared with linear protocol, fully fine-tuning is a much more common and prevalent transfer setting.
>
>
>
> **Q.6. Typos.**
>
> We have updated them in the manuscript.
>
> For Sinkhorn notation, our notation is equivalent to theirs. We use Trace notation for ease of understanding here.

---

> > ### Comment · Reviewer_Wv7b · 2022-11-24
> > **Thanks for the rebuttal**
> >
> > Thanks for the detailed response.
> > Most of my questions are addressed and answered. I would like to see this paper accepted.

---

> ### Author Response · Authors · 2022-11-17
> **Further comments and discussions will be appreciated!**
>
> Dear Reviewer Wv7b,
>
> We would like to thank you again for the detailed review. It would be appreciated if you share more of your thoughts based on our newly rebuttal. We are glad to address your further comments if there are any.
>
>
>
> In previous response,
>
> 1. We mainly discuss the difference between our work and Liu et al., (2022), both from methodology and experiment results. We also take our work as an interesting and non-trivial exploration on MAE with MoE, which is agreed by Reviewer uKwM and ycEm.
> 2. We explain the phenomenon of Fig 3(a), which is semantic-level collapse rather than token-level collapse.
> 3. The results of DeepClusterV2 and BYOL are different from (Liu et al., 2022) mainly because we adopt a more difficult and prevalent transfer setting, that we take the fine-tune protocol while (Liu et al., 2022) takes linear protocol.
> 4. We also add the missing part of MAE efficiency and fix some typos.
>
> We would appreciate it if you could kindly take a look at both the revision and our response to your comments. If you have any further questions, we are happy to discuss them!
>
>
>
> Best,
>
> Authors

---

> > ### Author Response · Authors · 2022-11-18
> > **Sincerely expecting further discussion**
> >
> > Dear Reviewer Wv7b,
> >
> > Given the end of the rebuttal period is less than 24 hours, we welcome any further comments and suggestions on our work from you to see if our responses solve your concerns.
> >
> > Thank you very much!
> >
> > Best,
> >
> > Authors

---

### Official Review · Reviewer_ycEm · 2022-10-26

**Confidence:** 5
**Clarity, Quality, Novelty And Reproducibility:** Generally, the paper is presented cle…
**Correctness:** 3
**Technical Novelty And Significance:** 4
**Empirical Novelty And Significance:** 2
**Recommendation:** 6

**Strength And Weaknesses:**

Generally, the paper is well-organized and easy to follow. The research topic is relatively new and interesting. The proposed MoCE seems correct and is well supported by the experimental results.
However, I also have the following concerns:
1. In the abstract and introduction, the authors position this work as a pretraining method. However, MoCE is based on a pre-trained MAE. Considering the extremely heavy cost of training foundation models, it would not be feasible.
2. Based on 1, the cost of training MoCE would be twice of MAE. To make a fair comparison, what if we train MAE with a dataset of twice scale or train MAE with more epochs?
3. If we focus on the negative transfer phenomenon, not combining MoE, there would be other transfer learning and model adaptation methods (e.g., prompt tuning) that could ease the negative transfer phenomenon. Could you compare it with some of them?
4. Since the input of Eq.5 is cluster centroid and there are only m clusters, is it enough to learn W_g well?
5. For Eq.6, is the loss implemented over all samples in the dataset or a batch?
In the experiment, only 11 small-scale classification tasks are evaluated, how about other more important tasks, e.g., object detection, semantic and instance segmentation?

**Summary Of The Paper:**

The paper studies the negative transfer phenomenon of MAE pre-trained models. Authors first experimentally show the existence of negative transfer and then show that natively using MoE can not solve the problem. Then, a clustering-based method is proposed to solve the problem.

**Summary Of The Review:**

I like the idea but believe the current experiments are not sufficient. If my concerns are solved, I will be willing to raise the score.


------ After rebuttal --------
In general, some of my concerns have been solved, e.g., training from scratch, and transferring to det and seg tasks. Thus, I would like to raise my score to borderline accept considering the interesting design of MOE.

However, I still have some concerns and hope the authors can address them in the final version. First, the added supervised pertaining results remind me that comparing MoCE with [1] would be necessary, which shows that simple MLPs can significantly improve the transferring performance of supervised pretraining.
Second, as you claim/argue the method is pretraining not transferring, it would be better to update all the results of MoCE with training from scratch instead of based on MAE to avoid confusion. This issue has also been pointed out by other reviewers (I believe it's feasible for the authors to train MoCE 1600 epochs from scratch since they already show some results about 200 epochs).

[1] Revisiting the Transferability of Supervised Pretraining: an MLP Perspective, CVPR 2022

---

> ### Author Response · Authors · 2022-11-14
> **Response to Reviewer ycEm**
>
> Thank you for your valuable and constructive reviews. Here we address your questions one by one.
>
> **Q.1. Feasibility of MoCE considering the cost of pre-training MAE as initialization.**
>
> R.1. Thanks for the comment. Actually, MoCE can be used without a pre-trained model. As an illustration, we train both MoCE and MAE from scratch for 200 epochs, referring to the following table. As can be seen, MoCE achieves significant improvement over 11 downstream datasets compared to MAE. On the other hand, it is a common practice to adopt open-source pre-training models to accelerate the training process[1,2]. With a pre-trained model, MoCE only needs 200 additional training epochs, which takes about 300 GPU hours, to produce 64 sub-models for task-customized pre-training of various downstream tasks. We think it is an acceptable budget.
>
>
>
> ▾ Table 1. Comparison of MAE and MoCE both training from scratch for 200 epochs.
>
> |      | Aircraft | Caltech | Cars  | C10   | C100  | DTD   | Flower | Food  | Pets  | SUN   | VOC   | Avg.  |
> | ---- | -------- | ------- | ----- | ----- | ----- | ----- | ------ | ----- | ----- | ----- | ----- | ----- |
> | MAE  | 64.73    | 85.91   | 77.10 | 92.92 | 72.50 | 73.30 | 93.11  | 73.14 | 88.70 | 57.84 | 73.27 | 77.50 |
> | MoCE | 71.16    | 90.55   | 82.46 | 96.06 | 76.56 | 74.57 | 95.70  | 79.67 | 92.58 | 62.20 | 84.25 | 82.34 |
>
>
>
> **Q.2. Cost of MoCE compared with MAE.**
>
> R.2. We clarify that based on a MAE pre-trained for 1600 epochs, we only train MoCE for another 200 epochs, instead of another 1600 epochs. For a fair comparison, we further train the 1600-epochs pre-trained MAE for another 200 epochs, which is denoted as **MAE*** in the last second row of Table 1 in the main paper. We can observe that MoCE also significantly outperforms **MAE*** with the same training cost.
>
>
>
> **Q.3. Comparison with other transfer learning and model adaptation methods (e.g., prompt tuning).**
>
> R.3. We clarify that methods of transfer learning and model adaptation, such as prompt tuning, is actually orthogonal to MoCE. Most of the transfer learning and model adaptation methods focus on the fast adaptation of a given pre-trained model, i.e., the fine-tuning stage. On the other hand, our method focuses on the pre-training stage and provides different pre-trained models for different downstream tasks. We believe that a combination of them will lead to better results, which will be investigated in our future work.
>
>
>
> **Q.4. Since the input of Eq.(5) is cluster centroid and there are only m clusters, is it enough to learn $W_g$ well?**
>
> R.4.  $W_g$ can be well learned because besides the cluster embedding matrix $C \in R^{m×D}$, $W_g$ is also optimized with respect to abundant input samples $x$. Here $m$ is the number of clusters and $D$ is the feature dimension. Specifically, the gradient of $W_g$ can be formulated as,
> $$
> \frac{\partial \mathcal{L}(x)}{\partial W_{g}} = \frac{\partial \mathcal{L}(x)}{\partial y} \cdot \frac{\partial y}{\partial G(x)} \cdot \frac{\partial G(x)}{\partial W_g} ,
> $$
> where $\mathcal{L}$ is the total loss, $x$ is the input image, $G(x)$ is the gate function as,
>
> $$
> G(x) = TopK(\sigma(\boldsymbol{W}_g \cdot Cluster\\_Emb(x))),
> $$
>
> and y is the output of our MoCE layer as,
> $$
> y = \sum_{i=1}^{N} G(\boldsymbol{x})_iE_i(\boldsymbol{x}).
> $$
> As in Eqn. 1, the first item $∂L(x)/∂y$ is the upstream gradient, and the second item $∂y/∂G(x) = E(x)$ is a function of $x$, which therefore will contribute to the gradient of $W_g$. For the last item $∂G(x)/∂W_g$, it is decided by the cluster embedding matrix $C$. Thus, $W_g$ is optimized with respect to input samples $x$ and clustering embedding matrix $C$ simultaneously.
>
>
>
> **Q.5. For Eq.(6), is the loss implemented over all samples in the dataset or a batch?**
>
> R.5. For Eq.(6), the loss is calculated over the samples in a batch in practise.
>
>
>
> **Q.6. In the experiment, only 11 small-scale classification tasks are evaluated, how about other more important tasks, e.g., object detection, semantic and instance segmentation?**
>
> R.6. As suggested, we have also provided the downstream semantic segmentation results on ADE20K in the revision. We can observe that MoCE also outperforms MAE. Due to the time limitation, other tasks will be added after the rebuttal period.
>
>
>
> ▾ Table 2. Semantic segmentation on ADE20K.
>
> |        | mIoU |
> | ------ | ---- |
> | DeiT   | 46.9 |
> | MoCov3 | 46.8 |
> | DINO   | 46.9 |
> | MAE    | 48.1 |
> | MoCE   | 48.3 |
>
>
>
> [1] Wu, Lemeng, et al. "Residual Mixture of Experts." ArXiv 2022.
>
> [2] Bai, Yutong, et al. "Masked Autoencoders Enable Efficient Knowledge Distillers." ArXiv 2022.

---

> ### Author Response · Authors · 2022-11-17
> **Further comments and discussions will be appreciated!**
>
> Dear Reviewer ycEm,
>
> Thank you for your valuable time to review our work and constructive feedbacks. We posted our response to your comments several days ago, and we wonder if you could kindly share some of your thoughts so we can keep the discussion rolling to address your concern if there are any.
>
> In previous response,
>
> 1. We mainly discuss the feasibility and cost of our method. Thanks to the open-source models, we are able to train MoCE in merely 300 GPU hours for 200 epochs to achieve our final results, which could be feasible and useful for the community to reproduce and follow.
> 2. We claim for the fairness of comparison in the main table, and difference with other methods like prompt tuning.
> 3. We explain the reason why W_g learns well, as each input x will contribute to the gradient of it.
> 4. As suggested, we provide the results of object detection, instance and semantic segmentation in the following table and Appendix A.5 in the revised paper. MoCE still achieves superior performance.
>
> |        | ADE      | COCO     |              |              |          |              |              |
> | ------ | -------- | -------- | ------------ | ------------ | -------- | ------------ | ------------ |
> |        | mIoU     | AP^{bb}  | AP^{bb}_{50} | AP^{bb}_{75} | AP^{mk}  | AP^{mk}_{50} | AP^{mk}_{75} |
> | DeiT   | 46.9     | 48.8     | 68.7         | 52.7         | 42.5     | 65.9         | 45.5         |
> | MoCov3 | 46.8     | 47.2     | 66.9         | 50.8         | 41.1     | 63.9         | 44.1         |
> | DINO   | 46.9     | 49.5     | 69.1         | 53.6         | 42.9     | 66.0         | 46.3         |
> | MAE    | 48.1     | 50.6     | 69.4         | 55.0         | 43.8     | 66.6         | 47.5         |
> | MoCE   | **48.3** | **51.1** | **69.8**     | **55.4**     | **44.2** | **67.0**     | **48.1**     |
>
>
>
> We would appreciate it if you could kindly take a look at both the revision and our response to your comments. If you have any further questions, we are happy to discuss them!
>
>
>
> Best,
>
> Authors

---

> > ### Comment · Reviewer_ycEm · 2022-11-17
> > **Thanks for the response**
> >
> > Thanks for the detailed response. In general, some of my concerns have been solved, e.g., training from scratch, and transferring to det and seg tasks. Thus, I would like to raise my score to borderline accept considering the interesting design of MOE.
> >
> > However, I still have some concerns and hope the authors can address them in the final version.
> > First, the added supervised pertaining results remind me that comparing MoCE with [1] would be necessary, which shows that simple MLPs can significantly improve the transferring performance of supervised pretraining.
> > Second, as you claim/argue the method is pretraining not transferring, it would be better to update all the results of MoCE with training from scratch instead of based on MAE to avoid confusion. This issue has also been pointed out by other reviewers (I believe it's feasible for the authors to train MoCE 1600 epochs from scratch since they already show some results about 200 epochs).
> >
> > [1] Revisiting the Transferability of Supervised Pretraining: an MLP Perspective, CVPR 2022

---

> > > ### Author Response · Authors · 2022-11-18
> > > **Response to Reviewer ycEM**
> > >
> > > Thanks for your support of our work. We are glad to have further discussion on your concerns here.
> > >
> > > **Q1. Comparison between MoCE and [1].**
> > >
> > > Thank you for pointing out this interesting work. [1] shows MLP is crucial for supervised learning and contrastive learning, which has not been explored in mask image modelling pre-training task, e.g., MAE and MoCE.
> > >
> > > We believe it is an orthogonal direction with MoCE and is glad to add more comparisons later.
> > >
> > > **Q2. MoCE with training from scratch is important.**
> > >
> > > We agree with your concerns on the importance of training from scratch, and we have provided the related results when pre-trained for 200 epochs. We will update the results of training MoCE for 1600 epochs in our final version due to the short rebuttal time.
> > >
> > > [1] Revisiting the Transferability of Supervised Pretraining: an MLP Perspective, CVPR 2022.
> > >
> > > [2] An empirical study of training self-supervised vision transformers, ArXiv 2021.

---

### Official Review · Reviewer_uKwM · 2022-10-31

**Confidence:** 4
**Correctness:** 3
**Technical Novelty And Significance:** 3
**Empirical Novelty And Significance:** 3
**Recommendation:** 8

**Clarity, Quality, Novelty And Reproducibility:**

Clarity: Overall the paper is a good read, and it is reasonably clear. One thing not clear to me: for data clustering, what features do it use? I am thinking the features are important, and if they are computed online, their FLOPs should also be counted? One could compute it offline, but it is still a computation budget over there if we want to compare overall pre-training cost fairly. Another (potentially minor) thing: for PSNR comparison in Fig 4, is it evaluating the reconstruction quality of the pre-training (auto-encoding) task? Or is it something else?

Quality: I think this is a useful exploration marrying MAE and MoE. The paper has good illustrations, nice storyline, and sufficient amount of experiments. One pity is that the MAE w/ MoE model is not pre-trained from scratch, and this may cause some noise in the signals, but the presented work is done with high-quality.

Novelty: While the techniques in the paper (how to do clustering, how to do gating in MoE) do appear to be existing ones, I think the work has directional novelty in exploring MAE w/ MoEs.

Reproducibility: If there is no difficult constraints, I hope the code of the paper is released to facilitate reproducing the results.

**Strength And Weaknesses:**

Strengths:
- This is one of the first works that I am aware to study MAE and MoE jointly. The exploration on this direction is interesting and of significance.
- The paper motivates the approach by pointing to an empirical result of MAE suffering from the negative transfer phenomenon, and the final approach (guided by the motivation) is able to significantly outperform the MAE baseline. Overall it looks like a healthy research cycle to me.

Weaknesses:
- One major concern I am having with MoE-kind approach is about larger models. While used in a sparse way, MoE is still having a lot of parameters, and if directly comparing against MAE of the same backbone, it does not sound too fair to me. So I would like to see two things: 1) with a larger MAE model that roughly has the same number of parameters as a model with MoE, what's the comparison? and 2) Whether the improvements of the current model can transfer to larger models. Related: in table 3, I am not able to find the column for MAE. All I find are columns for TokenMoE and MoCE, while the caption says MAE is under comparison.
- Related to the above, I am not sure the "negative transfer" phenomenon still exists with larger and larger MAE models? The hypothesis here is that maybe with larger models, it can capture both the man-made objects and the natural ones from ImageNet?
- Maybe it is demanding too much computation overhead, but I noticed that the method in the paper is built from pre-trained MAE, and not training from scratch.  So I am wondering what would MoCE be like when pre-trained from scratch?
- For the distillation loss, I am not sure why it still maintains an efficiency advantage over TokenMoE because the "whole" network is used as a "teacher", and presumably it can be costly to forward through all the parameters.
- (minor as I don't know where to put it in the review) Why does MoCo v3 achieve so good results on Table 1? Given that MoCo v3 is better than MAE, isn't it more reasonable to start MoE explorations from MoCo v3 given the results listed in the table? In the same table, I am also curious to see what the supervised ViT will achieve on these downstream datasets.


**Summary Of The Paper:**

The paper initiates an interesting exploration of MAE with mixture of experts (MoE). The method is quite well-motivated, with interesting and fairness-in-mind designs, and a good amount of experiments devoted to it. The final results are reported using a suite of 11 downstream classification tasks, typically used to evaluate self-supervised learning methods. The proposed method is shown to significantly improve the overall accuracy, while maintaining efficiency.

**Summary Of The Review:**

I would be on the acceptance side for the paper given the pros and cons listed above. I hope the authors can address the clarification questions I have, and try to tackle the empirical study (e.g. with larger models) as best as they can.

---

> ### Author Response · Authors · 2022-11-14
> **Response to Reviewer uKwM (1/2)**
>
> Thank you for your valuable and constructive reviews. Here we address your questions one by one.
>
> **Q.1-1. Experiments of MAE with a larger model whose model size is similar to that of MoCE for a fair comparison.**
>
> R.1-1. 1). The setting used in our work focuses on the fair comparison of **FLOPs**, referring to Table 3 in the main paper. Since TokenMoE and MoCE always activate only one expert throughout the whole pre-training and finetuning procedure, the FLOPs value is maintained close to MAE.
>
> 2). Apart from this criterion, as suggested, we further provide the comparison on equal **parameter** counts as suggested by the reviewer. As shown in the following table, we train MAE under same parameter count as the whole model of MoCE, and MoCE still outperforms MAE(MoCE size) consistently. Detailed discussion has been added in Appendix A.2 of the revised paper.
>
>
> ▾ Table 1. Comparison of MAE and MoCE under equal parameter counts.
>
> |                | #params | Aircraft | Caltech | Cars  |  C10  | C100  |  DTD  | Flower | Food  | Pets  |  SUN  |  VOC  | Avg.  |
> | :------------: | :-----: | :------: | :-----: | :---: | :---: | :---: | :---: | :----: | :---: | :---: | :---: | :---: | :---: |
> | MAE(MoCE size) | 178.03  |  74.43   |  90.30  | 85.50 | 96.90 | 83.80 | 74.84 | 96.30  | 81.86 | 92.97 | 62.98 | 85.51 | 84.13 |
> |      MoCE      | 178.03  |  78.73   |  90.61  | 88.56 | 97.79 | 84.68 | 74.04 | 96.94  | 86.24 | 93.07 | 65.05 | 85.26 | 85.54 |
>
>
>
> **Q.1-2. Whether the improvements of the current model can transfer to larger models.**
>
> R.1-2. Yes, the improvements of the current model can transfer to larger models. The following table shows the results of MoCE with ViT-Large ( 2.57× larger than ViT-Base), where MoCE (last row) still shows consistent improvement over MAE (first row), demonstrating its scalability. Detailed discussion has been added in Appendix A.3 of the revised paper.
>
>
>
> ▾ Table 2. Analysis on the larger architecture.
>
> |            | Aircraft | Caltech | Cars  |  C10  | C100  |  DTD  | Flower | Food  | Pets  |  SUN  |  VOC  |   Avg.    |
> | :--------: | :------: | :-----: | :---: | :---: | :---: | :---: | :----: | :---: | :---: | :---: | :---: | :-------: |
> |   MAE-IN   |  74.30   |  93.97  | 88.60 | 97.85 | 82.47 | 77.61 | 96.67  | 81.22 | 93.97 | 67.99 | 88.30 |   85.72   |
> | MAE-SplitA |  79.70   |  91.59  | 89.33 | 96.97 | 80.38 | 78.67 | 95.44  | 82.97 | 92.49 | 68.73 | 82.41 |   85.33   |
> | MAE-SplitB |  73.42   |  90.80  | 86.00 | 96.18 | 78.73 | 77.34 | 96.75  | 83.63 | 94.92 | 66.06 | 85.85 |   84.52   |
> |    MoCE    |  87.04   |  94.86  | 90.72 | 98.29 | 87.49 | 76.65 | 97.38  | 88.21 | 95.89 | 69.49 | 89.13 | **88.65** |
>
>
>
> **Q.1-3. Results of MAE in Table 3.**
>
> R.1-3. Thanks for your kind reminder and we have provided the results of MAE in Table 3 in the revised version. The results are also provided as follows.
>
>
>
> ▾ Table 3. Training (top 3 rows) and testing (last 3 rows) efficiency comparison.
>
> |           | MAE    | TokenMoE | MoCE   |
> | --------- | ------ | -------- | ------ |
> | Params(M) | 111.91 | 178.03   | 178.03 |
> | FLOPs(G)  | 9.80   | 9.81     | 9.81   |
> | Speed ↑   | 1.41x  | 1x       | 1.18x  |
> | Params(M) | 85.88  | 152.00   | 85.88  |
> | FLOPs(G)  | 16.88  | 16.88    | 16.88  |
> | Speed ↑   | 1.37x  | 1x       | 1.37x  |
>
>
>
> **Q.2. Will the negative transfer phenomenon still exist for large-scale models?**
>
> R.2. Yes, the negative transfer phenomenon still exists for large-scale models.
>
> The negative transfer phenomenon mainly exists when a common pre-trained model is used for various downstream tasks, due to the inevitable semantic gaps between the pre-training and downstream datasets, rather than the architecture.
>
> To verify, we further provide the results of MAE-Large trained on the two ImageNet-splits following the same setting of our preliminary experiments in the Sec. 3.1, referring to Table 2 in Q.1-2 . Observation can be made that MAE-Large still suffers from negative transfer problem, which our proposed method MoCE can successfully alleviate. We will include experiments of larger models (e.g., ViT-Huge) in the further revised version, due to the time limitation of the rebuttal period. Detailed discussion has been added in Appendix A.3 of the revised paper.

---

> > ### Author Response · Authors · 2022-11-14
> > **Response to Reviewer uKwM (2/2)**
> >
> > **Q.3. Performance of MoCE without pre-training.**
> >
> > R.3. It is a common practice to utilize pre-trained dense models as initialization to accelerate pre-training [1,2]. As suggested, we provide results of MoCE trained from scratch with 200 epochs in the following table. As can be seen, MoCE still outperforms MAE consistently in various downstream tasks. We have added the detailed discussion in Appendix A.4 of the revised paper.
> >
> >
> >
> > ▾ Table 4. Comparison of MAE and MoCE both training from scratch for 200 epochs.
> >
> > |      | Aircraft | Caltech | Cars  | C10   | C100  | DTD   | Flower | Food  | Pets  | SUN   | VOC   | Avg.  |
> > | ---- | -------- | ------- | ----- | ----- | ----- | ----- | ------ | ----- | ----- | ----- | ----- | ----- |
> > | MAE  | 64.73    | 85.91   | 77.10 | 92.92 | 72.50 | 73.30 | 93.11  | 73.14 | 88.70 | 57.84 | 73.27 | 77.50 |
> > | MoCE | 71.16    | 90.55   | 82.46 | 96.06 | 76.56 | 74.57 | 95.70  | 79.67 | 92.58 | 62.20 | 84.25 | 82.34 |
> >
> >
> >
> > **Q.4. Efficiency of the distillation loss.**
> >
> > R.4. We clarify that the extra time of introducing the distillation loss is negligible compared with pre-training since we only need to infer the network once an epoch. In practical, we store the features of the whole network at the end of each epoch, and use such features as guidance for distillation.
> >
> > Besides, this extra time has already been included when we compare speedup performance in Table 3 (Row ‘Speed’) in the main paper. We have updated the related explanation in the ‘Implementation details’ of Sec. 4.1 in the revised version.
> >
> >
> >
> > **Q.5-1. Why not choose MoCo-v3 as the baseline considering its promising performance?**
> >
> > R.5-1. We use MAE as a baseline mainly because its efficiency (i.e., more than 3× faster than MoCo-v3) and simplicity (i.e., not need to use Siamese architecture). On the other hand, the proposed MoCE can be naturally extended to MoCo-v3, which we will leave as future work.
> >
> >
> >
> > **Q.5-2. Performance of supervised ViT in Table 1.**
> >
> > R.5-2. As suggested, we provide the results of supervised ViT on downstream tasks in the following table and also Table 1 in the revised paper. As can be seen, our MoCE still obtains consistent improvement over the supervised baseline.
> >
> >
> >
> > ▾ Table 5. Comparison of supervised ViT and MoCE.
> >
> > |      | Aircraft | Caltech | Cars  | C10   | C100  | DTD   | Flower | Food  | Pets  | SUN   | VOC   | Avg.  |
> > | ---- | -------- | ------- | ----- | ----- | ----- | ----- | ------ | ----- | ----- | ----- | ----- | ----- |
> > | MAE  | 76.55    | 89.98   | 86.19 | 96.79 | 83.96 | 75.09 | 93.94  | 85.17 | 92.54 | 64.54 | 87.22 | 84.72 |
> > | MoCE | 78.73    | 90.61   | 88.56 | 97.79 | 84.68 | 74.04 | 96.94  | 86.24 | 93.07 | 65.05 | 85.26 | 85.54 |
> >
> >
> >
> > **Q.6. What features are used for data clustering? The computation cost of offline clustering should be considered for a fair comparison.**
> >
> > R.6. Thank you for pointing this out.
> >
> > 1). The features are computed offline from a pre-trained MAE.
> >
> > 2). The computation cost of the offline clustering has already been included in Table 3 (Row ‘Speed’) of the main paper.
> >
> > 3). Specifically, the clustering procedure requires only 3min20s on a single GPU, which is negligible compared to the pre-training procedure of MoCE of 10 GPU days.
> >
> > 4). We have updated the details for clustering in Appendix A.1.
> >
> >
> >
> > **Q.7. For PSNR comparison in Fig 4, is it evaluating the reconstruction quality of the pre-training (auto-encoding) task? Or is it something else?**
> >
> > R.7. Yes, it is evaluating the reconstruction quality of the pre-training task, as you understand.
> >
> >
> >
> > [1] Wu, Lemeng, et al. "Residual Mixture of Experts." ArXiv 2022.
> >
> > [2] Bai, Yutong, et al. "Masked Autoencoders Enable Efficient Knowledge Distillers." ArXiv 2022.

---

### Author Response · Authors · 2022-11-14
**To All Reviewers**

We thank all the reviewers for their time, insightful suggestions, and valuable comments. We are glad that ALL reviewers regard our topic as important and interesting, and has directional novelty in exploring MAE w/MoEs (reviewer ’uKwM’), with good experimental results (reviewer ’uKwM’, ’Wv7b’ and ’Emir’ ), well-supported analysis and visualization (reviewer ’ycEm’ and ’Wv7b’).

We respond to each reviewer’s comments in detail below. We have also revised the main paper and appendix according to the reviewers’ suggestions.

The main changes are listed as follows:

1. In Table 1, we add comparisons with supervised ViT.
2. In Table 2, we add the experimental results of SDR(Liu et al., (2022)), and provide the related analysis in Sec. 4.2.
3. We update the efficiency statistics of MAE in Table 3.
4. As suggested by Reviewer1 and 4, we add the cost analysis of clustering in Appendix A.1.
5. As suggested by Reviewer1, we added the comparison between MAE and MoE under the same parameter counts in Appendix A.2.
6. As suggested by Reviewer1, we add the MoCE for larger architecture (ViT-L) in Appendix A.3.
7. We add the performance of MoCE without pre-training in Appendix A.4 following the suggestion of Reviewer 1 and 2.
8. We add the results on other downstream tasks, e.g., detection, instance and segmentation tasks in Appendix A.5, as suggested by the Reviewer 2.

Note that we marked the revisions in blue. We hope that our efforts can address the reviewers’ concerns well.

Best,

Authors

---

### Author Response · Authors · 2022-11-17
**Further comments and discussions are appreciated!**

Dear all,

We are grateful for the valuable comments and suggestions given by the reviewers, which help clarify our work. We upload a revised version of the manuscript, and try to address the concerns of the reviewers.

We appreciate your further comments on our work since the end of rebuttal time is approaching!



Best,

Authors

---

### Decision · Program_Chairs · 2023-01-20

**Decision:**

Accept: notable-top-25%

**Justification For Why Not Higher Score:**

The current paper addresses an important problem, negative transfer of pre-trained models to downstream tasks. The paper follows a healthy cycle of research, as mentioned by one of the senior reviewers, that pinpoints true research problem followed by a systematic study based on well-established baselines and benchmarks. Therefore, AC believes this paper shall be highlighted as a spotlight at the conference. However, an oral is not appropriate, not only because of the limited quota of orals, but also because the current methodology is mainly built upon MAE and MoE, which is not so pronounced compared to MAE, MoCo, and SimCLR.

**Justification For Why Not Lower Score:**

All reviewers agree to accept this paper unanimously, some with favorable thoughts of the paper. To make a fair decision, AC finds no evidence to assign a lower score for this good paper.

**Metareview: Summary, Strengths And Weaknesses:**

This paper improves the Masked Auto-encoders (MAE) with a Mixture of Cluster-conditional Experts (MoCE) approach. The key finding of this paper is that the semantically irrelevant pre-training information might result in negative transfer, even for larger models pre-trained with bigger datasets. Note that negative transfer is a very important problem for pre-trained models, and this paper tackles this problem with a reasonable solution -- introducing the clustering information into the MoE-based backbone. Evaluation shows significant empirical benefit compared with the close baselines MAE and TokenMAE.

The paper initially received four insightful and constructive review reports, though of mixed ratings. Authors made a big effort in the rebuttal phase, by providing updated version with most requested results (larger models, fair model size, ablations, etc.) augmented to the paper and the response letters. Reviewers actively interacted with the authors, acknowledged the updates, and converged finally to unanimous positive recommendation.

AC concurs with the four reviewers' opinion on the paper. Authors are requested to incorporate all rebuttal material into the future version, in particular, add train-from-scratch results which were not provided in the rebuttal phase due to time and resource constraint.

**Note From Pc:**

if the above contains the word "oral" or "spotlight" please see: "oral" presentation means -> notable-top-5% and "spotlight" means -> notable-top-25%. As stated in our emails, we are disassociating presentation type from AC recommendations